# Comparison of the Effects of Sugammadex and Pyridostigmine on Postoperative Nausea and Vomiting and the Recovery Profile in Pediatric Patients Undergoing Strabismus Surgery: A Prospective, Double-Blind, Observational Study

**DOI:** 10.3390/medicina61101826

**Published:** 2025-10-12

**Authors:** Se Hun Kim, Hwa Song Jong, Eun Gyo Ha, Su Yeon Cho, Ki Tae Jung, Dong Joon Kim

**Affiliations:** 1Department of Anesthesiology and Pain Medicine, Chosun University Hospital, Gwangju 61453, Republic of Korea; sepal1116@gmail.com (S.H.K.); yddizy@chosun.ac.kr (H.S.J.); gkdmsry2004@naver.com (E.G.H.); isycho@chosun.ac.kr (S.Y.C.); mdmole@chosun.ac.kr (K.T.J.); 2Department of Anesthesiology and Pain Medicine, College of Medicine and Medical School, Chosun University, Gwangju 61452, Republic of Korea

**Keywords:** postoperative vomiting, nausea, neuromuscular blockage reversal, sugammadex, pyridostigmine

## Abstract

*Background and Objectives*: Postoperative nausea and vomiting (PONV) is a common and potentially crucial side effect in pediatric patients. Neuromuscular blockade reversal drugs (NMBRDs) used during surgery have been associated with PONV. This study investigated whether sugammadex, a recently approved NMBRD for children in Korea, induces PONV and possible changes after NMBRD administration in children undergoing strabismus surgery. *Materials and Methods*: In total, 60 pediatric patients (3–16 years old) undergoing strabismus surgery with general anesthesia were included. They were divided into two groups: sugammadex (group S, *n* = 30) or pyridostigmine (group P, *n* = 30). The primary endpoint was the incidence of PONV using the Baxter Animated Retching Faces (BARF) scale at 0.5, 1, 3, and 6 h after NMBRD administration. The secondary endpoints included the recovery time (train-of-four > 0.9) and changes in heart rate following NMBRD administration. *Results*: There was no significant difference in PONV incidence between the groups according to the BARF scale (13.3% vs. 13.3% at 0.5 h, 6.7% vs. 3.3% at 1 h). Sugammadex demonstrated a significantly faster recovery time than pyridostigmine (*p* < 0.001). The changes in heart rate were more significant in the sugammadex group than those in the pyridostigmine group after NMBRD administration (*p* < 0.001); however, the heart rate returned to preoperative levels without any need for rescue medications or anticholinergics during the emergence period. *Conclusions*: There was no significant difference in PONV incidence between the administration of sugammadex and pyridostigmine in pediatric patients after strabismus surgery. Nevertheless, sugammadex appeared to facilitate faster recovery from the neuromuscular blockade without requiring intervention for the heart rate.

## 1. Introduction

Children more commonly aspirate stomach contents perioperatively, and when postoperative nausea and vomiting (PONV) occurs after surgery, it may be more difficult to manage than in adults. They are especially vulnerable to prolonged fasting, which can also result in complications during the postoperative recovery process. This condition is recognized for exhibiting potentially fatal adverse effects, which include dehydration, electrolyte imbalance, and impaired wound healing. These complications can delay the surgical recovery and discharge periods while also causing considerable discomfort that may lead to long-term adverse emotional outcomes for pediatric patients [1]. Appropriate perioperative fasting time and preoperative pharmacologic intervention are commonly used as prophylaxis of PONV, as they are essential for general anesthesia surgery [2]. However, these drugs can have side effects, such as arrhythmias, gastrointestinal movement reduction, and extrapyramidal symptoms [3].

Until recently, conventional neuromuscular blockade reversal drugs (NMBRDs) such as neostigmine or pyridostigmine were the only available options for reversing neuromuscular blockade (NMB) in pediatric surgery. Now, sugammadex has emerged as an alternative for children, following a decade of safe use in adults, with reports suggesting a reduced risk of postoperative nausea and vomiting (PONV) [4]. Similarly, studies in adults have shown that a traditional reversal drug, pyridostigmine, is associated with a lower incidence of PONV compared to neostigmine [5]. Conventional neuromuscular blockade (NMB) reversal agents, such as neostigmine and pyridostigmine (acetylcholinesterase inhibitors), cause undesirable muscarinic side effects by increasing acetylcholine (ACh). This necessitates co-administration of an anticholinergic to manage effects such as bradycardia. Critically, this heightened ACh activity stimulates GI motility and increases gastric secretions, which can lead to clinical symptoms such as abdominal cramping, diarrhea, and a subsequent increased likelihood of PONV [6]. Although several studies have reported a comparison of the PONV occurrence between neostigmine and sugammadex [4], limited data exist specifically for children.

Considering the potential benefits of sugammadex and the limited data on its effects in children, particularly regarding PONV, this study aimed to compare the effects of sugammadex alone and a pyridostigmine–glycopyrrolate combination on the incidence of PONV after pediatric strabismus surgery. We hypothesized that sugammadex would offer a better outcome in terms of PONV, with a faster recovery from neuromuscular blockade and without complications.

## 2. Materials and Methods

### 2.1. Study Design and Ethical Considerations

The Institutional Review Board of Chosun University Hospital, Republic of Korea, approved this randomized, prospective, double-blind, and observational study (2022-006-001; approval date, 12 October 2022). Informed consent was obtained from each patient’s legal guardian. This study was performed in compliance with the ethical standards of the Declaration of Helsinki (2013 revision).

### 2.2. Inclusion and Exclusion Criteria

This study included all patients who underwent general anesthesia for strabismus surgery at Chosun University Hospital from January to July 2023, provided they met the specific inclusion and exclusion criteria. We excluded patients who were outside the 3-to-16-year age range, had comorbidities of an American Society of Anesthesiologists Physical Status (ASA-PS) of III or higher, presented with congenital anomalies, had symptoms of a recent upper respiratory infection (within two weeks), or suffered from asthma or other respiratory tract diseases.

The patients were randomly divided into two groups using a computer-generated random number table, according to the use of the NMBRD, which was administered at the end of general anesthesia (Figure 1). Group S (*n* = 30) received 4.0 mg/kg of sugammadex (Bridion^®^, MSD, Seoul, Republic of Korea), and Group P (*n* = 30) received 0.2 mg/kg of pyridostigmine (Pygmin, Hana Pharm, Seoul, Republic of Korea) with 0.01 mg/kg of glycopyrrolate (Tabinul, Hana Pharm, Seoul, Republic of Korea). The anesthesia nurse prepared the drugs, sugammadex and pyridostigmine with glycopyrrolate, in a 5 ml syringe, blindly. The nurse did not participate in the study room’s anesthesia. Additionally, an anesthesiologist other than the one who obtained informed consent from the study population group administered anesthesia to the patient.

### 2.3. Anesthesia, NMBRDs, PONV, Heart Rate and Covariates

Anesthesia was induced with 5 mg/kg thiopental sodium (Pentothal sodium; JW Pharma, Seoul, Republic of Korea), and 2 vol% sevoflurane (Sevofran; Hana Pharmal, Seoul, Republic of Korea) was administered with 5 L/min O_2_ through a face mask. After confirmation of loss of consciousness, 0.6 mg/kg of rocuronium bromide (Esmeron^®^; MSD Korea, Seoul, Republic of Korea) was administered for muscle relaxation [7,8]. Neuromuscular quantitative monitoring was performed to the adductor pollicis muscle using a train-of-four (TOF) electromyograph-based neuromuscular transmission monitoring (E-NMT, GE healthcare^®^, Needham, MA, USA).

At the end of the surgery, the inhalation agent was discontinued, and we supplied 5 L/min of oxygen for the removal of the remaining sevoflurane, a possible risk factor of PONV. When the second twitch occurred, the NMB was reversed with the assigned NMBRD by conventional administration timing for patient safety: sugammadex or pyridostigmine with glycopyrrolate. Endotracheal tube extubation was performed when the TOF ratio was greater than 0.9, a normal breathing pattern had returned, and the face was no longer flaccid. After extubating and confirming a normal breathing pattern, the patient was transferred to the post-anesthesia care unit (PACU).

In the PACU, the children were kept with their parents. The NMBRD type used was blinded to the nurses. The incidence of nausea was assessed using the Baxter Animated Retching Faces (BARF) scale [9]. In this study, PONV was defined as nausea or vomiting within 6 h of surgery. Attempting to vomit was considered nausea; antiemetics were not given routinely but only in the event of vomiting. The postoperative heart rate was determined minimally 30 min after the admission to PACU via ECG monitoring.

Variable and outcome measurements:(1)Age, gender, height, weight, body mass index, ASA-PS, and anesthesia time.(2)The primary endpoint was the frequency estimation of PONV using the BARF scale—PONV between 30 min and 1 h after surgery was measured in the PACU and between 3 and 6 h after surgery in the ward.(3)The secondary endpoints were the recovery time to TOF > 0.9 from the NMBRD administration, change in heart rate (HR) after NMBRD administration, and the difference between the basal and maximal change in the HR after NMBRD administration, as measured by an anesthetist blinded to the NMBRD.

### 2.4. Sample Size Calculation and Statistical Analysis

The effect size was 0.5, which was a large effect size suggested by Cohen’s conventional criteria by the previous studies on POV after use of sugammadex in pediatric strabismus surgery patients [8]. Using α = 0.05 with a power of 80%, the total sample size was calculated to be 52. After assuming a 15% drop out rate, 30 patients were allocated to each group.

Statistical analyses were performed using GraphPad Prism (v.10.5.0, GraphPad Software) and Statistical Package for the Social Sciences (SPSS, v. 27.0, IBM, Armonk, NY, USA). Continuous variables are presented as the mean ± standard deviation, and categorical variables are presented as the number of patients (%). The normality of the distribution was evaluated using the Shapiro–Wilk test (or Kolmogorov–Smirnov test). All data were found to be normally distributed. The Chi-squared test (or Fisher’s exact test) was used to compare the incidence of PONV using the BARF scale, which was the primary goal of this study. The Chi-squared test (or Fisher’s exact test) was used to analyze sex and the class ASA-PS. *T*-tests were used to analyze the age, body mass index (BMI), and anesthesia time. Changes in HR were analyzed via repeated measures two-way analysis of variance, and differences between groups were analyzed via *t*-test. Tukey’s honestly significant difference test was used for post hoc testing. Statistical significance was defined as a *p* value < 0.05. The BMI was categorized according to the WHO classification.

## 3. Results

### 3.1. Patient Characteristics

A total of 60 patients were assessed for eligibility and enrolled in this study (Figure 1). There were no significant differences in the baseline characteristics between the two groups (Table 1).

### 3.2. The Incidences of PONV

The primary endpoint, PONV incidence, assessed using the BARF scale, was not significantly different between the two groups (Table 2).

### 3.3. The Recovery Profile of Sugammadex and Pyridostigmine

Regarding the secondary endpoints, the recovery time to TOF > 0.9 was significantly faster in group S, and the changes in HR were significantly greater in group S than those in group P (Table 3). Heart rate variation (%) represents the changes in the heart rate before and after NMBRD injection. Since the HR management required varies by age, we needed to identify age-appropriate correlations. The HR change limit is the percentage limit according to age and race (nonwhite and black) (5th/2nd) [10,11] by ages accordingly.

In terms of the changes in HR, except for the period immediately following NMBRD administration (*p* = 0.018), no significant differences in heart rate were observed between the two groups at any other measurement point (Table 4). Although it showed a minimal HR over the limit of the fifth percentile in either group, no rescue HR control medications were required, due to being within the race-specific second percentile limit of HR during the emergence, and the HR recovered spontaneously (Figure 2). The following data are presented in Figure 2: personal records of the maximal point of HR change by group. The *p* value was borderline; so age and sex, as well as the allowance of limitations of spontaneous resolution for rescue intervention, were considered. Female children demonstrated a substantial decrease in HR trend by age in Group S (R^2^ = 0.5826, *p* = 0.0024).

## 4. Discussion

Sugammadex, a gamma-cyclodextrin, works by encapsulating and deactivating steroidal NMBRDs, and it has the strongest affinity for the most widely used of these drugs [12]. The connection between NMBRDs and PONV outcomes has been hypothesized, with suggestions that reversal drugs may affect the emetic center and that PONV could be caused by muscarinic receptor effects [13,14]. Similarly, co-administering glycopyrrolate with pyridostigmine has been shown to potentially prevent PONV after laparoscopic surgery and Cesarean sections under neuraxial anesthesia [15]. Nonetheless, sugammadex used on its own may be more effective than pyridostigmine at preventing PONV in patients using opioid-based intravenous patient-controlled analgesia [14].

Recent meta-analyses of NMBRDs have reported that sugammadex holds potential advantages over neostigmine, similar to those of pyridostigmine in the rate of PONV following general anesthesia (OR = 0.64, [0.46–0.90]) [16]. However, there is a lack of meta-analysis on the effects of sugammadex on PONV in pediatric patients, as its use in this population is a recent development. Additionally, studies have produced conflicting results regarding the effects of NMBRDs on neuromuscular blockade.

Based on prior adult studies, we investigated the incidence of PONV up to 6 h after surgery in our trial. For instance, research on ENT surgery found that patients who received sugammadex had a significantly lower PONV incidence (3%) than those given a neostigmine–atropine mixture (20%) at 6 h post-surgery (*p* = 0.013) [17].

PONV is a complex multifactorial problem in pediatric patients. The Eberhart score helps predict the risk of postoperative vomiting by considering a patient’s age, surgery time, family history, and whether they are undergoing strabismus surgery [18]. Strabismus has been specifically identified as a significant risk factor for PONV [18]. This risk score demonstrates that as the risk factors accumulate, the incidence of vomiting increases [19]. Accordingly, some studies have created risk scores specifically for predicting vomiting in children [20]. For PONV, patients with an Apfel simple score of 2 or more are candidates for intraoperative management [21]. The best approach to managing PONV in children is a personalized one, tailored to the patient’s specific risk factors and the surgical procedure. This may also include the use of antiemetics, such as selective serotonin receptor (5-HT_3_) antagonists, local anesthetics, or non-pharmacological interventions, such as acupuncture or acupressure [1].

Previous research [22] supports the use of sugammadex at a dose of 4 mg/kg for the reversal of moderate and deep rocuronium- and vecuronium-induced neuromuscular blockade in patients aged 2–17 years, as it showed no significant differences in clinically relevant bradycardia, hypersensitivity, or anaphylaxis compared to neostigmine. Based on these findings, we conducted our study on a pediatric population ranging from 3 to 16 years of age in Korean population. Figure 2 shows that the females in Group S had a statistically significant decrease in minimal HR. However, such results require measurement in a larger experimental group, as the data from one older female child may act as a confounding variable. Nevertheless, overall, the minimal HR tended to decrease with increasing age in children. Consequently, when administering sugammadex to relatively older children, we need to pay attention to changes in heart rate. However, in this experiment, it did not exceed the second percentile during recovery, and it recovered naturally. Recent studies report that, while there is no gender difference in PONV before puberty, after puberty, PONV increases approximately threefold in women [23,24]. While there is potential for bradycardia associated with sugammadex, particularly as a patient’s minimum HR decreases with age, this risk did not necessitate intervention, according to the results of our study. This finding suggests that the clinical benefit of reducing a significant source of PONV compared to neostigmine in a high-risk population may exceed the mild severity and self-resolving risk of bradycardia observed in the present cohort in Korean pediatrics. Nevertheless, due to the limited sample size of this study (*n* = 60), a direct causal relationship between the drug’s use and HR changes in specific age and gender subgroups could not be definitively established. Consequently, further research with larger patient cohorts and different ethnicity is necessary to specifically investigate the heart rate profile of sugammadex in relation to age and sex, thereby enabling the development of more tailored and safer anesthetic protocols. In pediatric patients with congenital anomalies or cardiac disease or those who developed bronchospasm and oxygen desaturation with bradycardia following endotracheal intubation, serious adverse events such as cardiac arrest due to severe sugammadex-induced hypotension have been reported. Therefore, this drug should be used with caution in this population [25,26].

Because of the difficulty in objectively assessing nausea in children, vomiting is often adopted as a more objective clinical endpoint for managing both postoperative and post-discharge nausea. The BARF provides a reliable and valid solution for this. This pictorial scale uses five cartoon faces (ranging from happy to a child vomiting) and asks children to select the face that best represents how they feel. This tool has proven its utility in clinical trials and practice for assessing the efficacy of antiemetics and other PONV interventions. For pediatric patients aged 6 years or older, a BARF score of 4 or more is considered an indication for rescue antiemetics, with a minimum clinically important difference of 1.47 [9].

Several limitations were identified in this trial. Primarily, the small sample size hinders the generalizability of the results, underscoring the need for a larger sample for a more robust analysis [27] and regarding the statistical inaccuracy of using Cohen’s d for sample size calculation with a binary outcome (PONV incidence). We based our calculation on the unusually large effect size (a 26.7% absolute risk reduction) from the prior work [6], but this proved risky, as underscored by our finding of a 0% difference. Since our previous studies failed to identify differences after those observed during emergence, a larger sample size for the experimental group may have been necessary. Therefore, our findings should be interpreted not as a failed comparison of superiority but as a preliminary confirmation of non-inferiority. The similarity in PONV rates between sugammadex monotherapy and the conventional regimen is of clinical significance. This result aligns with adult data showing no difference in PONV rates between sugammadex and pyridostigmine (OR = 0.95; *p* = 0.281) [5]. Its use is supported by the fact that it is at least non-inferior to the conventional standard regimen. To ensure an accurate comparison, future research should match patients based on specific surgical details, such as the muscles affected in strabismus or whether the procedure is monocular versus binocular. Prior research indicates that children frequently experience clinically significant nausea that goes untreated if not accompanied by vomiting [9]. In addition, all pediatric patients received gastric decompression (GD) immediately after endotracheal intubation to protect the gastric contents from regurgitation due to unintended gas insufflation during mask ventilation for anesthesia induction [8], which is a procedure known to prevent POV in children; hence, the true incidence of POV might have been underestimated. As GD is independently known to effectively reduce the baseline POV risk in children, this pre-emptive intervention substantially minimized the overall PONV incidence in both groups. Consequently, this study’s power was insufficient to detect any minor incremental difference between the two NMB reversal agents. Our findings should be interpreted as a crucial confirmation of non-inferiority. The long-term effects of pyridostigmine, particularly on post-discharge nausea and vomiting, also need to be explored due to its prolonged duration of action. Different timings of drug administration could influence the heart rate before and after the NMBRD injection. Sugammadex is often given more quickly at the end of surgery, a time when residual volatile anesthetics still exert an effect [28]. Conversely, conventional NMBRD is administered when the train-of-four (TOF) count reaches 2 or higher. This represents an area for potential optimization in future studies. The comparison of sugammadex alone to a conventional drug introduces a confounding variable, as glycopyrrolate is known to possess antiemetic properties. This may have masked a true difference in PONV incidence with pyridostigmine, the primary reversal agent. We acknowledge this limitation, which is inherent in comparing sugammadex to the clinically mandated combination of anticholinesterase with an anticholinergic. Future research should consider a trial comparing sugammadex alone versus pyridostigmine alone or sugammadex + glycopyrrolate versus pyridostigmine + glycopyrrolate to isolate the antiemetic effect of the NMBRDs. Further investigation is needed to determine the heart rate-related aspects of interpretation. This investigation should include a comparison of the same period after drug administration and a larger number of patients. This will help to further investigate the effects of the drug on age and gender. Although no treatment for bradycardia was needed, the potential for combining sugammadex and glycopyrrolate in future trials is worth considering for similar conditions with bradycardia [28]. Additionally, a separate trial is already underway to investigate whether glycopyrrolate alone can prevent PONV, though its focus is on opioid use and not the surgical procedure itself [29].

Pyridostigmine, unlike neostigmine, does not significantly increase the risk of PONV in adults [5]. However, its potential side effects when co-administered with anticholinergics must be considered, necessitating its judicious use within a comprehensive anesthetic regimen. As an integral part of general anesthesia, there are several drugs used in surgery that can directly or indirectly cause PONV. The use of sugammadex in pediatric patients is a relatively recent development; so, additional research is needed to establish its efficacy and safety. As with all medications, healthcare providers must carefully evaluate the risks and benefits of sugammadex and communicate these to patients and their families. Many anesthetic drugs used during surgery can directly or indirectly lead to PONV. Given the lack of substitutes for these agents and their long duration record of safe use, it is challenging to justify a change simply to mitigate PONV. Therefore, the value of recent studies demonstrated that sugammadex reduces the risk of PONV in adults could be even more significant in children, a hypothesis that warrants further investigation.

## 5. Conclusions

The incidence of PONV was not significantly different between the sugammadex and pyridostigmine groups following pediatric strabismus surgery. This finding suggests that sugammadex is non-inferior to the conventional regimen in terms of PONV safety in this pediatric population. But sugammadex has been reported to cause side effects like bradycardia and hypotension in pediatric patients with congenital heart disease. Sugammadex is a viable alternative to conventional NMBRDs for pediatric patients undergoing strabismus surgery without co-morbid diseases, such as congenital heart disease. This alternative provides a rapid, predictable reversal while simplifying the anesthetic regimen by eliminating the need for routine anticholinergic co-administration.

## Figures and Tables

**Figure 1 medicina-61-01826-f001:**
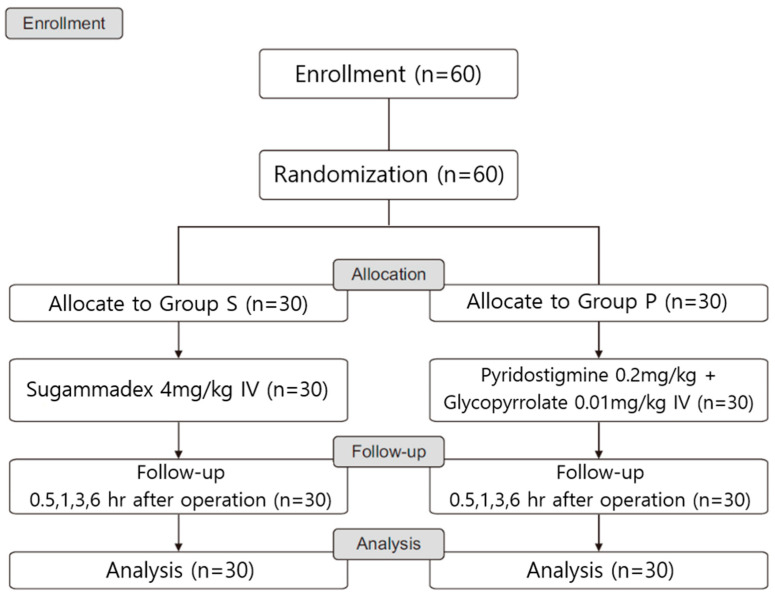
Study flow diagram. Enrollment of 60 pediatric patients (3 to 16 years old) undergoing strabismus surgery (American Society of Anesthesiologists physical status I–II) and their randomization into two groups according to the NMBRD used: Group S received 4.0 mg/kg sugammadex; Group P received 0.2 mg/kg pyridostigmine with 0.01 mg/kg glycopyrrolate.

**Figure 2 medicina-61-01826-f002:**
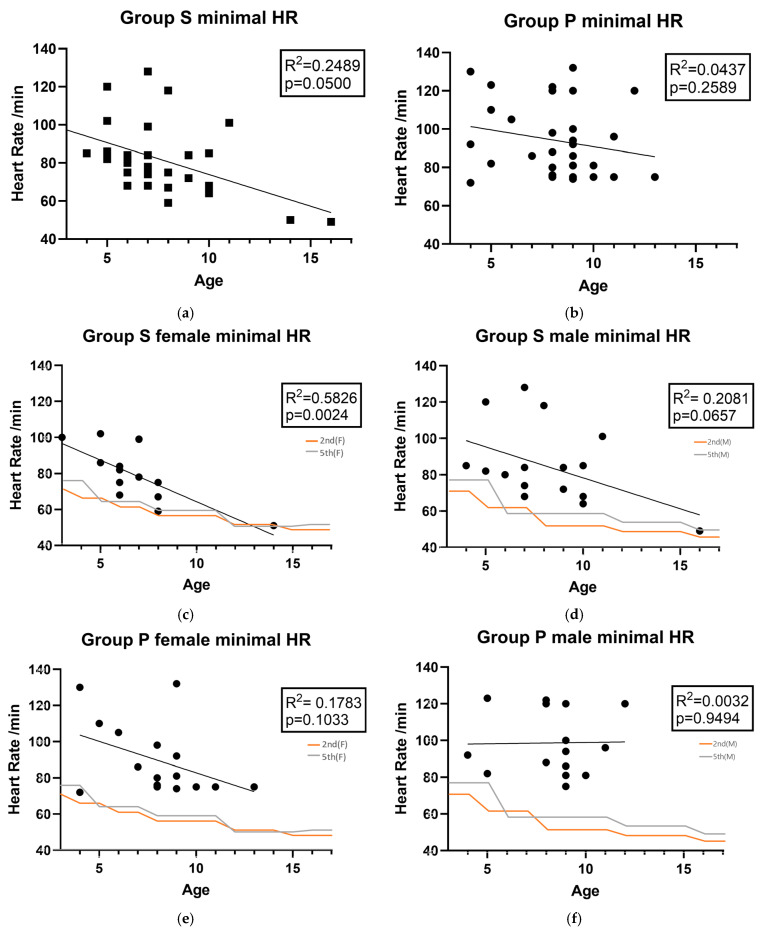
Minimal HR after NMBRD use for each individual and its correlation with sex and age: (**a**) Group S minimal HR, (**b**) Group P minimal HR, (**c**) Group S female minimal HR, (**d**) Group S male minimal HR, (**e**) Group P female minimal HR, (**f**) Group P male minimal HR; Group S, sugammadex; Group P, pyridostigmine bromide with glycopyrrolate. HR, heart rate; NMBRD, neuromuscular blocking reversal drug; 2nd (F), mean second percentile (female), 5th (F), mean fifth percentile (female) 2nd (M), mean second percentile (male), 5th (M), mean fifth percentile (male).

**Table 1 medicina-61-01826-t001:** Demographic data.

Variable	Group S (*n* = 30)	Group P (*n* = 30)	*p*-Value
Age (years)	7.60 ± 2.8 (6.5–8.7)	8.07 ± 2.3 (7.2–8.9)	0.484
Sex (M/F)	(17/13)	(14/16)	0.605
BMI	18.9 ± 3.9 (17.4–20.4)	18.3 ± 4.8 (16.5–20.1)	0.592
ASA-PS Class (I/II)	(28/2)	(26/4)	0.671
Anesthesia Time (min)	64.3 ± 12.2 (59.8–68.9)	62.9 ± 11.2 (62.9–71.3)	0.364

The values are presented as the mean ± standard deviation (SD), (95% confidence interval (CI)), or number. Statistical significance is set at *p* < 0.05. Group S, sugammadex; group P, pyridostigmine + glycopyrrolate; ASA-PS: American Society of Anesthesiologists physical status; BMI: body mass index.

**Table 2 medicina-61-01826-t002:** The incidences of PONV assessed over time using the BARF scale.

Time After Emergence	Group S (*n* = 30)	Group P (*n* = 30)	*p*-Value
0.5 h	4/30 (13.3%)	4/30 (13.3%)	1.000
1 h	2/30 (6.7%)	1/30 (3.3%)	0.554
3 h	0	0	-
6 h	0	0	-

Values are presented as the BARF score over 4/total with incidence (%). Group S, sugammadex; Group P, pyridostigmine + glycopyrrolate; PONV, postoperative nausea and vomiting; BARF, Baxter Animated Retching Faces.

**Table 3 medicina-61-01826-t003:** Changes upon recovery from anesthesia.

Variables	Group S (*n* = 30)	Group P (*n* = 30)	*p*-Value
TOF > 0.9 time (s)	62.7 ± 27.0 (50.9–74.5)	125.9 ± 47.6 (105.3–147.1)	<0.001
Heart rate variation (%)	13.9 ± 11.2	0.3 ± 11.7	<0.001
Changes over-limit HR (*n*)	(3/0)	(1/0)	

Values are presented as the mean ± SD (95% CI). Group S, sugammadex; Group P, pyridostigmine bromide with glycopyrrolate. TOF, train-of-four; HR change limit normal percentile according to age and race in nonwhite and black (5th/2nd).

**Table 4 medicina-61-01826-t004:** Changes in heart rate over time.

Changes in HR	Group S (*n* = 30)	Group P (*n* = 30)	*p*-Value
Baseline (before induction)	100.1 ± 15.7 (94.3–106.0)	102.5 ± 11.2 (98.3–106.7)	0.504
Before HR NMBRD use	95.1 ± 17.9 (88.4–101.8)	95.0 ± 16.9 (88.7–101.3)	0.994
After HR NMBRD use	81.8 ± 19.2 (74.7–89.0)	94.0 ± 19.3 (86.8–101.2)	0.018
PACU (30 min after NMBRD)	107.0 ± 15.0 (101.4–112.6)	103.1 ± 10.1 (99.3–106.8)	0.240

The values are presented as the mean ± SD (95% CI). Group S, sugammadex; Group P, pyridostigmine bromide with glycopyrrolate. HR, heart rate; NMBRD, neuromuscular blocking reversal drug; PACU, post-anesthesia care unit.

## Data Availability

The datasets analyzed used in this study are available from the corresponding author upon reasonable request.

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
