# Peer review of "Comparison of the Effects of Sugammadex and Pyridostigmine on Postoperative Nausea and Vomiting and the Recovery Profile in Pediatric Patients Undergoing Strabismus Surgery: A Prospective, Double-Blind, Observational Study"

_medicina, 2025, doi:10.3390/medicina61101826_

Round 1
Reviewer 1 Report
Comments and Suggestions for Authors
The authors present a prospective, double-blind, randomized study comparing sugammadex and pyridostigmine on PONV in pediatric patients undergoing strabismus surgery. The topic is clinically relevant. However, there are some issues needed to be addressed.
Major concerns
In the Methods section, the authors stated that the sample size was calculated using Cohen's conventional effect size of 0.5. This is an error as Cohen’s d is a metric for a study comparing the continuous outcomes. In the current study the outcome is binary; hence, the effect size should be defined as an odds ratio or absolute difference in proportions. The cited study [cit 6] founds a very large effect: a 26.7% absolute risk reduction in POV (30% vs. 3.3%, OR=0.08). This is unusual for clinical studies. Basing the calculation of sample size on the assumption of such a large effect is risky. The current study's finding of a 0% difference underscores the variability of PONV and suggests the prior effect size may not be generalizable.
The comparison of sugammadex to pyridostigmine + glycopyrrolate is a limitation. Glycopyrrolate is an anticholinergic agent with known antiemetic properties. It likely suppresses the rate of PONV in the group P. That may introduce bias towards null hypothesis. The potential benefits of sugammadex in that case may be masked by the effect of glycopyrrolate.
Minor Points
The use of gastric decompression in all patients is a known intervention to reduce POV. In the current study it lowers down the event rate making it extremely difficult to detect a difference between the groups.
The interpretation of the statistically significant heart rate change after sugammadex administration is too brief.
Several statements in the Introduction and Discussion require citations to support their claims:
Lines 42,48 (The statement that perioperative fasting is "essential for surgery"), line 55 (the claim that sugammadex has a "reduced risk of PONV" is central to the rationale of the study but is currently unsupported)
Lines 59-60 Consider rephrasing to make it clear for the reader that neostigmine/pyridostigmine cause muscarinic effects and adding a citation on how the acetylcholinesterase inhibitors affect gastrointestinal motility.
Lines 257,266,272,279, 283 Citations needed to support claims.
Author Response
|
Comments 1: In the Methods section, the authors stated that the sample size was calculated using Cohen's conventional effect size of 0.5. This is an error as Cohen’s d is a metric for a study comparing the continuous outcomes. In the current study the outcome is binary; hence, the effect size should be defined as an odds ratio or absolute difference in proportions. The cited study [cit 6] founds a very large effect: a 26.7% absolute risk reduction in POV (30% vs. 3.3%, OR=0.08). This is unusual for clinical studies. Basing the calculation of sample size on the assumption of such a large effect is risky. The current study's finding of a 0% difference underscores the variability of PONV and suggests the prior effect size may not be generalizable.
|
|
Response 1: We appreciate the reviewer's point out regarding the use of Cohen’s d for sample size calculation in a study with a binary primary outcome (PONV incidence). We acknowledge that, from a strictly statistical perspective, the effect size should ideally be expressed as an odds ratio or an absolute difference in proportions. It was hypothesized that sufficient odds ratios would be measured during the planning stage, and accordingly, the above experiment was conducted. However, it must be acknowledged that the experiment is subject to limitations in terms of equivalence, as no absolute difference in the odd ratio was observed, irrespective of the underlying cause.
1) The experiments considered as prior research investigated the effects of physical interventions (gastric decompression). This intervention is physical in nature, thereby distinguishing it from the drug usage difference observed in this experiment. This discrepancy should have been examined with greater precision. However, given the absence of relevant prior research, the research team conducting this experiment established the research methodology by considering the extant literature [cit. 6].
2) The fact that vomiting differences were only observed during emergence in [Cit 6] above, while results 30 minutes later, such as in the PACU, showed relatively no difference, may be recognized as a statistical limitation in the experimental design, as this aspect was not sufficiently considered.
3) It is important to note that all patients in the current trial received Gastric Decompression (GD). This pre-emptive intervention, known to prevent POV in children, served as a baseline control to eliminate a major source of PONV risk before the NMBRD comparison began. We kindly ask for your understanding that this pre-emptive intervention is a routine procedure at our hospital to ensure safe general anesthesia for all pediatric patients. It was necessary to perform it to obtain IRB approval, which requires ensuring patient safety and no differences between children. We fully acknowledge the limitation that, to achieve a more precise comparison between drugs or interventions, the comparison should have been made without performing GD.
4) Although it did not show a dramatic difference in incidence rates compared to situations like physical GD in postoperative vomiting, this suggests that, given the contextual difference in measurement methods requiring children to be conscious enough to recognize pictures, and at least in terms of PONV, sugammadex monotherapy in pediatric patients routinely receiving GD may demonstrate safety comparable to the combination therapy of pyridostigmine plus glycopyrrolate—which is considered superior to conventional neostigmine use for PONV management. This confirms a potentially pivotal finding: in pediatric patients, as in adults, it is at least non-inferior to existing therapy. In adult study, [Cit.5] shows PONV were low in the sugammadex group (odds ratio, 0.65; 95% confidence interval, 0.59–0.72; p < 0.0001) and pyridostigmine group (odds ratio, 0.22; 95% confidence interval, 0.20–0.24; p < 0.0001) compared to the neostigmine group, while there was no difference between sugammadex and pyridostigmine (odds ratio, 0.95; 95% confidence interval, 0.86–1.04; p = 0.281).
This confirms the critical finding that sugammadex is at least non-inferior to the conventional pyridostigmine + glycopyrrolate regimen in terms of PONV in pediatric patients undergoing strabismus surgery with general anesthesia.
We add this comment in discussion part. 278-289 "And regarding the statistical inaccuracy of using Cohen's d for sample size calculation with a binary outcome (PONV incidence). We based our calculation on the unusually large effect size (a 26.7% absolute risk reduction) from the prior work [6], but this proved risky, as underscored by our finding of a 0% difference. This pre-emptive intervention substantially minimized the overall PONV incidence in both groups, as GD is independently known to effectively reduce baseline POV risk in children. As a result, the study's power was not enough to identify any small, gradual differences between the two NMB reversal agents. Therefore, our findings should be interpreted not as a failed comparison of superiority, but as a crucial confirmation of non-inferiority. The similarity in PONV rates between Sugammadex monotherapy and the conventional Pyridostigmine + Glycopyrrolate regimen is of clinical significance. This result aligns with adult data showing no difference in PONV rates between Sugammadex and Pyridostigmine (OR=0.95; p=0.281) [5]. Demonstrating that it is at least non-inferior to the conventional standard regimen supports its use."
|
|
Comments 2: The comparison of sugammadex to pyridostigmine + glycopyrrolate is a limitation. Glycopyrrolate is an anticholinergic agent with known antiemetic properties. It likely suppresses the rate of PONV in the group P. That may introduce bias towards null hypothesis. The potential benefits of sugammadex in that case may be masked by the effect of glycopyrrolate. |
|
Response 2: We totally understand the reviewer's concern regarding the limitations of glycopyrrolate's properties. While we were aware of glycopyrrolate's antiemetic effect when designing the experiment, we sought to compare it in the actual clinical setting where it is routinely used at our hospital. Additionally, considering the safety implications of bradycardia associated with the combined use of pyridostigmine and other anticholinergics, we contemplated adding atropine. However, we lacked routine experience with combined atropine use and opted for glycopyrrolate due to concerns about atropine's blood-brain barrier (BBB) penetration and potential for fever. We regret that this aspect may have interfered with obtaining precise experimental results. Furthermore, since sugammadex is not routinely combined with glycopyrrolate in clinical practice, pyridostigmine was not used in the experiment. To clearly state these limitations, we will add the following statement to the discussion section. 311-317 "The comparison of sugammadex alone to a conventional combination drug (pyridostigmine + glycopyrrolate) introduces a confounding variable, as glycopyrrolate is known to possess antiemetic properties. This may have masked a true difference in PONV incidence between pyridostigmine, the primary reversal agents. We acknowledge this limitation, which is inherent in comparing sugammadex to the clinically mandated combination of anticholinesterase with an anticholinergic. Future research should consider a trial comparing sugammadex alone versus pyridostigmine alone (if ethically feasible and safe) or sugammadex + glycopyrrolate versus pyridostigmine + glycopyrrolate to isolate the antiemetic effect of the NMBRDs” |
|
Comments 3: The use of gastric decompression in all patients is a known intervention to reduce POV. In the current study it lowers down the event rate making it extremely difficult to detect a difference between the groups.
Response 3: We totally agree reviewer's opinion that and we believe the dramatic reduction in GD POV may have diminished the discriminatory power of this study. However, it demonstrates non-inferiority as an intervention under identical conditions for drug use, indicating the drug can be safely used alone, which is meaningful. Additional wording has been inserted regarding this aspect. 297-302 " All patients received Gastric Decompression (GD). As GD is independently known to effectively reduce baseline POV risk in children, this pre-emptive intervention substantially minimized the overall PONV incidence in both groups. Consequently, the study's power was insufficient to detect any minor, incremental difference between the two NMB reversal agents. Our findings need to be interpreted not as a failed comparison of superiority, but as a crucial confirmation of non-inferiority."
Comments 4 : The interpretation of the statistically significant heart rate change after sugammadex administration is too brief.
Response 4: We totally agree that the statistical interpretation related to heart rate changes is relatively simple. However, this interpretation is limited to the aspect that sugammadex monotherapy can be used safely without excessive heart rate depression, even without concomitant glycopyrrolate administration. Due to the relatively small experimental group size, differences based on gender and age were observed during the experiment, and only the presentation of related phenomena was conducted. The following limitation is presented: additional interpretation is needed regarding heart rate-related aspects. (317-321) "Further investigation is needed to determine the heart rate-related aspects of interpretation. This investigation should include a comparison of the same period after drug administration and a larger number of patients. This will help to further investigate the effects of the drug on age and gender. "
Comments 5 : Several statements in the Introduction and Discussion require citations to support their claims: Response 5: Lines 42,48 (The statement that perioperative fasting is "essential for surgery"), : Preoperative fasting is essential for general anesthesia in children, so the phrase has been revised to “essential for general anesthesia for protection of aspiration.” Since excessive fasting can adversely affect PONV, appropriate preoperative fasting is essential for general anesthesia surgery, and adhering to an appropriate fasting period can reduce the incidence of PONV. (49-51) “Essential for general anesthesia surgery, appropriate perioperative fasting time and preoperative pharmacologic intervention are commonly used to prophylaxis of PONV [2]” and added the citation [2].
Comments 6 : line 55 (the claim that sugammadex has a "reduced risk of PONV" is central to the rationale of the study but is currently unsupported) Response 6: (58) Based on the above points, the following citation has been added [4]. Now, sugammadex has emerged as an alternative for children, following a decade of safe use in adults, with reports suggesting a reduced risk of postoperative nausea and vomiting (PONV) [4].
Comments 7: Lines 59-60 Consider rephrasing to make it clear for the reader that neostigmine/pyridostigmine cause muscarinic effects and adding a citation on how the acetylcholinesterase inhibitors affect gastrointestinal motility. Response 7: Based on the above points, we have rephrased the text to make it clearer for the reader and added the citation as follows. The potential occurrence of bradycardia and other undesirable muscarinic, a side effect associated with conventional acetylcholinesterase inhibitors, such as neostigmine, necessitates concomitant use of anticholinergics. Consequently, it may induce muscarinic side effects like the potential incidence of less gastrointestinal movement which may increase the likelihood of PONV occurrence. to (60-66) "Conventional neuromuscular blockade (NMB) reversal agents, such as neostigmine and pyridostigmine (acetylcholinesterase inhibitors), cause undesirable muscarinic side effects by increasing acetylcholine (ACh). This necessitates co-administration of an anticholinergic to manage effects like bradycardia. Critically, this heightened ACh activity stimulates GI motility and increases gastric secretions, which can lead to clinical symptoms like abdominal cramping, diarrhea, and a subsequent increased likelihood of PONV [6]."
Comments 8 Lines 257,266,272,279, 283 Citations needed to support claims. : We have subdivided the citations and arranged them accordingly, and have provided additional citations. Response 8 257 : [28] 278 266 : [9] 296 272 : [29] 307 279 : [29] 323 283 : Pyridostigmine, unlike neostigmine, doesn't significantly increase the risk of PONV in pediatrics. Since data in children is limited, we will modify it for adults and present the citation [5]. (326-327) |

Reviewer 2 Report
Comments and Suggestions for Authors
In the manuscript Comparison of the Effects of Sugammadex and Pyridostigmine on Postoperative Nausea and Vomiting and Recovery Profile in Pediatric Patients Undergoing Strabismus Surgery: A Prospective, Double-Blind, Observational Study, the authors provide a comparison between the use of two different drugs for the treatment of postoperative nausea and vomiting in pediatric patients that underwent strabismus surgery. The study appears well written, following a good protocol regarding the rules of scientific research and writing and providing personal experience regarding the use of new drugs in preventing postoperative complications in a sensible population such as children. The study has recent and relevant references and the authors should be congratulated for their work. Prior to being acceptable for publication there is a major issue that the authors need to address:
Rows 236-244
The figure 2 showed Group S minimal HR decreased by aging in female only. This suggests that when bradycardia is observed in females after puberty, intervention related to heart rate may be necessary when administering sugammadex.
Please consider rephrasing for clarity by aging in female only. Same for intervention related to heart rate.
However, in this experiment, it did not exceed the 2nd percentile during recovery without rescue medication and recovered naturally. Recent studies report that while there is no gender difference in PONV before puberty, after puberty, PONV increases approximately threefold in women. [21,22] This highlights that sugammadex, which demonstrates a significant reduction in PONV compared to neostigmine, should be considered in the selection of NMBRD for post-pubertal female, who represent a relative high-risk group for PONV.
In the 1st part of this paragraph the authors claim that under sugammadex, bradycardia appears in post-pubescent female patients and that heart rate should be monitored to avoid bradychardia, while in the 2nd part the authors claim these patients to have more frequent PONV and thus, sugammadex being better suited for them. The authors should consider making a decision regarding wether the use of this drug is an advantage or a disadvantage and clarify this aspect in the manuscript.
Furthermore, the authors state that this occurs in postpubescent female patients. Have the authors analyzed whether the patients were pre, post or undergoing puberty? If not, these speculations should not be correlated with the authors findings.
There are existing reports of bradychardia in children under suggamadex regardless of gender and at young ages and the authors should address that caution should be exerted in detecting such heart rate changes, especially since these modifications are in the fda references(https://www.accessdata.fda.gov/drugsatfda_docs/label/2024/022225s014lbl.pdf):
Carvalho EVG, Caldas SMC, Costa DFPPMD, Gomes CMGP. Bradycardia in a pediatric population after sugammadex administration: case series. Braz J Anesthesiol. 2023;73(1):101-103. doi:10.1016/j.bjane.2021.12.011
Vaswani ZG, Smith SM, Zapata A, Gottlieb EA, Sheeran PW. Bradycardic Arrest in a Child with Complex Congenital Heart Disease Due to Sugammadex Administration. J Pediatr Pharmacol Ther. 2023;28(7):667-670. doi:10.5863/1551-6776-28.7.667
Author Response
|
Comments and Suggestions for Authors In the manuscript Comparison of the Effects of Sugammadex and Pyridostigmine on Postoperative Nausea and Vomiting and Recovery Profile in Pediatric Patients Undergoing Strabismus Surgery: A Prospective, Double-Blind, Observational Study, the authors provide a comparison between the use of two different drugs for the treatment of postoperative nausea and vomiting in pediatric patients that underwent strabismus surgery. The study appears well written, following a good protocol regarding the rules of scientific research and writing and providing personal experience regarding the use of new drugs in preventing postoperative complications in a sensible population such as children. The study has recent and relevant references and the authors should be congratulated for their work. Prior to being acceptable for publication there is a major issue that the authors need to address:
Comments 1: Rows 236-244 The figure 2 showed Group S minimal HR decreased by aging in female only. This suggests that when bradycardia is observed in females after puberty, intervention related to heart rate may be necessary when administering sugammadex.
Please consider rephrasing for clarity by aging in female only. Same for intervention related to heart rate.
|
|
Response 1: We are in full agreement with the reviewer's perspective. Direct interpretation should be cautious, as the finding that HR decreases only in women could be due to an outlier in the sole female participant and the relatively older age of one individual. Therefore, we will revise the above statement as follows: "The figure 2 showed Group S minimal HR decreased by aging in female only. This suggests that when bradycardia is observed in females after puberty, intervention re-lated to heart rate may be necessary when administering sugammadex."
to
(243-248) " The figure 2 showed Group S, female children showed a statistically significant decrease in minimal HR. However, such results require measurement in a larger experimental group as the data from one older female child may act as a confounding variable. Nevertheless, overall, minimal HR tended to decrease with increasing age in children. Consequently, when administering sugammadex to relatively older children, need to pay attention to changes in heart rate." |
|
Comments 2: However, in this experiment, it did not exceed the 2nd percentile during recovery without rescue medication and recovered naturally. Recent studies report that while there is no gender difference in PONV before puberty, after puberty, PONV increases approximately threefold in women. [21,22] This highlights that sugammadex, which demonstrates a significant reduction in PONV compared to neostigmine, should be considered in the selection of NMBRD for post-pubertal female, who represent a relative high-risk group for PONV.
In the 1st part of this paragraph the authors claim that under sugammadex, bradycardia appears in post-pubescent female patients and that heart rate should be monitored to avoid bradychardia, while in the 2nd part the authors claim these patients to have more frequent PONV and thus, sugammadex being better suited for them. The authors should consider making a decision regarding wether the use of this drug is an advantage or a disadvantage and clarify this aspect in the manuscript.
Furthermore, the authors state that this occurs in postpubescent female patients. Have the authors analyzed whether the patients were pre, post or undergoing puberty? If not, these speculations should not be correlated with the authors findings. |
|
Response 2: : We fully agree with the reviewer's opinion. It is difficult for this study to determine whether any drug is advantageous for a specific gender, particularly females, at a specific time period after puberty. Therefore, we will modify the content as follows "This highlights that sugammadex, which demonstrates a significant reduction in PONV compared to neostigmine, should be considered in the selection of NMBRD for post-pubertal female, who represent a relative high-risk group for PONV."
to
(250-261) "While there is potential for bradycardia associated with sugammadex, particularly as a patient's minimum HR decreases with age, this risk did not necessitate intervention, according to the results of our study. This finding suggests that the clinical benefit of reducing a significant source of PONV compared to neostigmine in a high-risk population may exceed the mild severity, self-resolving risk of bradycardia observed in the present cohort. Nevertheless, due to the limited sample size of this study (n = 60), a direct causal relationship between the drug's use and HR changes in specific age and gender subgroups could not be definitively established. Consequently, further research with larger patient cohorts is necessary to specifically investigate the heart rate profile of sugammadex in relation to age and sex, thereby enabling the development of more tailored and safer anesthetic protocols."
Comments 3: There are existing reports of bradycardia in children under sugammadex regardless of gender and at young ages and the authors should address that caution should be exerted in detecting such heart rate changes, especially since these modifications are in the fda references(https://www.accessdata.fda.gov/drugsatfda_docs/label/2024/022225s014lbl.pdf):
Carvalho EVG, Caldas SMC, Costa DFPPMD, Gomes CMGP. Bradycardia in a pediatric population after sugammadex administration: case series. Braz J Anesthesiol. 2023;73(1):101-103. doi:10.1016/j.bjane.2021.12.011
Vaswani ZG, Smith SM, Zapata A, Gottlieb EA, Sheeran PW. Bradycardic Arrest in a Child with Complex Congenital Heart Disease Due to Sugammadex Administration. J Pediatr Pharmacol Ther. 2023;28(7):667-670. doi:10.5863/1551-6776-28.7.667
|
|
Response 3: : We totally Agree. Based on the reviewer's comments, we will add the following statement and citation
(261-266) “In pediatric patients with congenital anomalies or cardiac disease, or those who developed bronchospasm and oxygen desaturation with bradycardia following endotracheal intubation, serious adverse events such as cardiac arrest due to severe sugammadex-induced hypotension have been reported. Therefore, this drug should be used with caution. [26,27]" |

Reviewer 3 Report
Comments and Suggestions for Authors
In their prospective, double-blind, randomized study, the authors investigated whether sugammadex induces PONV and possible changes after NMBRD administration in 60 patients undergoing strabismus surgery. They concluded that there was no significant difference in PONV incidence between the administration of sugammadex and pyridostigmine in pediatric patients after strabismus surgery. Nevertheless, sugammadex appeared to facilitate faster recovery from the neuromuscular blockade without clinically relevant need for intervention for heart rate. The manuscript is generally well written, methodologically sound, and of interest to anesthesiologists and pediatric ophthalmologists. The randomized, prospective, double-blind design adds credibility, although there are limitations related to potential confounders and the interpretation of outcomes. I have several objections and suggestions for improvement.
- The introduction emphasizes the significance of PONV in pediatric surgery and notes the limited data on sugammadex use in children. However, the novelty is somewhat underplayed; although several adult studies exist, the pediatric context should be more convincingly justified.
- Although described as "randomized, prospective, double-blind," the methods section does not clearly explain the randomization process. The authors should precisely specify how randomization was carried out, as it is unclear from the methodology.
- In addition to the previous comment, the authors should explain how doctors and nurses were blinded, who prepared the drugs, and how they were presented to end users.
- Adding glycopyrrolate to the pyridostigmine group introduces a confounding factor because glycopyrrolate itself may affect PONV. This should be acknowledged as a limitation.
- The sample size calculation is explained, but the assumed effect size (0.5) might be too optimistic, increasing the chance of a type II error. This should be more clearly acknowledged as a limitation. Additionally, a post-hoc analysis should be performed to verify the achieved sample size.
- The finding that sugammadex enables faster recovery aligns with previous research, but the heart rate results are somewhat complex and should be interpreted more carefully. The subgroup analysis by age and sex (noting that female children show a decrease in HR with age) is interesting but underexplored and could be overinterpreted due to the small sample size.
- The discussion offers valuable context but could be more concise. Several points are repeated (e.g., role of glycopyrrolate, BARF scale utility).
- The limitations section is well developed but should more clearly highlight the small sample size and potential underestimation of PONV due to gastric decompression.
- Conclusions should be more cautious; stating that sugammadex is a "viable alternative" is reasonable, but claiming it is superior in PONV management is not supported.
Author Response
|
3. Point-by-point response to Comments and Suggestions for Authors Comments and Suggestions for Authors In their prospective, double-blind, randomized study, the authors investigated whether sugammadex induces PONV and possible changes after NMBRD administration in 60 patients undergoing strabismus surgery. They concluded that there was no significant difference in PONV incidence between the administration of sugammadex and pyridostigmine in pediatric patients after strabismus surgery. Nevertheless, sugammadex appeared to facilitate faster recovery from the neuromuscular blockade without clinically relevant need for intervention for heart rate. The manuscript is generally well written, methodologically sound, and of interest to anesthesiologists and pediatric ophthalmologists. The randomized, prospective, double-blind design adds credibility, although there are limitations related to potential confounders and the interpretation of outcomes. I have several objections and suggestions for improvement.
: Your commentary is greatly appreciated. I am genuinely grateful for the expressions of gratitude received. |
|
Comments 1: The introduction emphasizes the significance of PONV in pediatric surgery and notes the limited data on sugammadex use in children. However, the novelty is somewhat underplayed; although several adult studies exist, the pediatric context should be more convincingly justified |
|
Response 1 : We fully agree with the reviewer's opinion. we change the contents in introduction part
“ Postoperative nausea and vomiting (PONV) is a common and unpleasant experience for patients undergoing strabismus surgery under general anesthesia and PONV has been reported to occur more commonly in children than in adults. The condition is recognized for exhibiting potentially fatal adverse effects, which include dehydration, electrolyte imbalance, and impaired wound healing. These complications can delay surgical recovery and discharge, while also causing considerable discomfort that may lead to long-term adverse emotional outcomes for pediatric patients [1]. "
to
(42-53) " Children are relatively easier aspiration of stomach contents, and when vomiting occurs, Postoperative nausea and vomiting (PONV) may be more difficult to manage than in adults. They are especially vulnerable to prolonged fasting, which can also result in complications during the postoperative recovery process. The condition is recognized for exhibiting potentially fatal adverse effects, which include dehydration, electrolyte imbalance, and impaired wound healing. These complications can delay surgical recovery and discharge, while also causing considerable discomfort that may lead to long-term adverse emotional outcomes for pediatric patients [1]. Essential for general anesthesia surgery, appropriate perioperative fasting time and preoperative pharmacologic intervention are commonly used to prophylaxis of PONV [2]. However, these drugs can have side effects, like arrhythmias, gastrointestinal movement reduction and extrapyramidal symptoms [3]. "
|
|
Comments 2 & 3 : 2. Although described as "randomized, prospective, double-blind," the methods section does not clearly explain the randomization process. The authors should precisely specify how randomization was carried out, as it is unclear from the methodology. 3. In addition to the previous comment, the authors should explain how doctors and nurses were blinded, who prepared the drugs, and how they were presented to end users.
|
|
Response 2 & 3: "All the NMBRDs were prepared in a volume of total 4 mL in 5mL syringe by a nurse blinded to the experimental parameters."
to
(90-99) "The patients were randomly divided into two groups using a computer-generated random number table. The anesthesia nurse prepared the drugs, sugammadex and pyridostigmine with glycopyrrolate, in a 5ml syringe, blindly. The nurse cannot participate in the study room's anesthesia. Additionally, an anesthesiologist other than the one who obtained informed consent from the study population group administered anesthesia to the patient."
Comments 4: Adding glycopyrrolate to the pyridostigmine group introduces a confounding factor because glycopyrrolate itself may affect PONV. This should be acknowledged as a limitation.
Response 4 : We totally understand the reviewer's concern regarding the limitations of glycopyrrolate's properties. While we were aware of glycopyrrolate's antiemetic effect when designing the experiment, we sought to compare it in the actual clinical setting where it is routinely used at our hospital. Additionally, considering the safety implications of bradycardia associated with the combined use of pyridostigmine and other anticholinergics, we contemplated adding atropine. However, we lacked routine experience with combined atropine use and opted for glycopyrrolate due to concerns about atropine's blood-brain barrier (BBB) penetration and potential for fever. We regret that this aspect may have interfered with obtaining precise experimental results. Furthermore, since sugammadex is not routinely combined with glycopyrrolate in clinical practice, pyridostigmine was not used in the experiment. To clearly state these limitations, we will add the following statement to the discussion section.
(310-317) "The comparison of sugammadex alone to a conventional combination drug (pyridostigmine + glycopyrrolate) introduces a confounding variable, as glycopyrrolate is known to possess antiemetic properties. This may have masked a true difference in PONV incidence between pyridostigmine, the primary reversal agents. We acknowledge this limitation, which is inherent in comparing sugammadex to the clinically mandated combination of anticholinesterase with an anticholinergic. Future research should consider a trial comparing sugammadex alone versus pyridostigmine alone (if ethically feasible and safe) or sugammadex + glycopyrrolate versus pyridostigmine + glycopyrrolate to isolate the antiemetic effect of the NMBRDs"
Comments 5: The sample size calculation is explained, but the assumed effect size (0.5) might be too optimistic, increasing the chance of a type II error. This should be more clearly acknowledged as a limitation. Additionally, a post-hoc analysis should be performed to verify the achieved sample size.
Response 5 : We fully agree with the reviewer's opinion. As noted in the comments, the effect size clearly has limitations. Nevertheless, although our research team proceeded based on results from our prior studies, the limited number of relevant prior studies available during the research planning stage ultimately led to statistically constrained results. We acknowledge this limitation in the discussion section and kindly request your understanding that the results must be interpreted as preliminary findings or as evidence from a non-inferiority test. We added about this limitation in discussion part (281-285) "Since our previous studies failed to identify differences after those observed during emergence, a larger sample size for the experimental group may have been necessary. Therefore, our findings should be interpreted not as a failed comparison of superiority, but as a preliminary confirmation of non-inferiority"
Comments 6: The finding that sugammadex enables faster recovery aligns with previous research, but the heart rate results are somewhat complex and should be interpreted more carefully. The subgroup analysis by age and sex (noting that female children show a decrease in HR with age) is interesting but underexplored and could be overinterpreted due to the small sample size.
Response 6: We are in full agreement with the reviewer's perspective. Direct interpretation should be cautious, as the finding that HR decreases only in women could be due to an outlier in the sole female participant and the relatively older age of one individual. Therefore, we will revise the above statement as follows: "The figure 2 showed Group S minimal HR decreased by aging in female only. This suggests that when bradycardia is observed in females after puberty, intervention related to heart rate may be necessary when administering sugammadex."
to (243-249) " The figure 2 showed Group S, female children showed a statistically significant decrease in minimal HR. However, such results require measurement in a larger experimental group as the data from one older female child may act as a confounding variable. Nevertheless, overall, minimal HR tended to decrease with increasing age in children. Consequently, when administering sugammadex to relatively older children, we need to pay attention to changes in heart rate."
Comments 7: The discussion offers valuable context but could be more concise. Several points are repeated (e.g., role of glycopyrrolate, BARF scale utility).
Response 7: We try to simplify the discussion part the role of glycopyrrolate and BARF scale. 1) Erase this part : The study's exclusive use of the BARF scale also presents a challenge, as it may not fully capture the burden of PONV. 2) Glycopyrrolate's role was erased that universally known thing in discussion part. : Because glycopyrrolate's antiemetic role is revealed in the discussion section, we tried to shorten the contents about the drug, but despite the reviewer's comment, insufficient editing was done. Please accept my apologies for the insufficient shortening.
Comments 8: The limitations section is well developed but should more clearly highlight the small sample size and potential underestimation of PONV due to gastric decompression.
Response 8: : We agree this comment, and added about this (297-302) " All patients received Gastric Decompression (GD). As GD is independently known to effectively reduce baseline POV risk in children, this pre-emptive intervention substantially minimized the overall PONV incidence in both groups. Consequently, the study's power was insufficient to detect any minor, incremental difference between the two NMB reversal agents. Our findings need to be interpreted not as a failed comparison of superiority, but as a crucial confirmation of non-inferiority."
Comments 9: Conclusions should be more cautious; stating that sugammadex is a "viable alternative" is reasonable, but claiming it is superior in PONV management is not supported.
Response 9: : We fully agree. I have revised it to mean that it does not show superiority in the incidence of PONV but rather serves as a viable alternative in certain situations due to its relatively faster recovery and the absence of the need for concomitant anticholinergic therapy.
"The incidence of PONV was not significantly different between the groups that received sugammadex and pyridostigmine after pediatric strabismus surgery. Sugammadex was found to facilitate a more rapid recovery from the neuromuscular blockade when compared to the pyridostigmine, and it did not require routine use of anticholinergics and clinically significant interventions for heart rate control. Therefore, sugammadex offers a viable alternative to conventional anticholinesterase with anticholinergics, which have a long history of use in adults and have not yet been shown to improve PONV." to (340-348) "The incidence of PONV was not significantly different between the sugammadex and pyridostigmine groups following pediatric strabismus surgery. This finding suggests that Sugammadex is non-inferior to the conventional regimen in terms of PONV safety in this pediatric population. But sugammadex has been reported to cause side effects like bradycardia and hypotension in pediatric patients with congenital heart disease. Sugammadex is a viable alternative to conventional NMBRDs for pediatric patients undergoing strabismus surgery without co-morbid disease, such as congenital heart disease. This alternative provides a rapid, predictable reversal while simplifying the anesthetic regimen by eliminating the need for routine anticholinergic co-administration" |

Round 2
Reviewer 1 Report
Comments and Suggestions for Authors
All my comments have been addressed
Author Response
Thank you so much!!
Reviewer 2 Report
Comments and Suggestions for Authors
As a suggestion the authors could consider adding the ethnic background for their studied population as it may provide data regarding the applicability of their findings in different populations
Author Response
1. As a suggestion the authors could consider adding the ethnic background for their studied population as it may provide data regarding the applicability of their findings in different populations
: We agree the reviewer`s mention about the ethnic background consideration. We tried to check the limitation about mininal heart rate for safety check. So the value about of percentile could be limited by the ethnicity in asian polulation. So we added the mention about ethnicity in disscussion part for further evaluation for different populations.
Reviewer 3 Report
Comments and Suggestions for Authors
The authors addressed my comments thoroughly and enhanced the manuscript. The paper can be accepted in preset form.
Author Response
Thank you so much.